# DSP-EntCD: A Knowledge-Freezing, Entropy-Guided Remote Sensing Change Detection Network with Domain-Specific Pretraining

## Abstract

Remote sensing change detection refers to identifying surface changes in very-high-resolution images acquired over the same geographic area at different times. It serves as a core technology in natural resource supervision and intelligent urban management. However, in most real-world scenarios, the changed regions occupy only a small portion of the image, causing existing methods to be biased toward background detection. In addition, change detection faces the challenge of spatio-temporal multi-scale heterogeneity, where change targets exhibit significant scale variations across temporal sequences and spatial dimensions, increasing the difficulty of feature modeling. To address these issues, we propose a knowledge-freezing two-stage training framework, termed Domain-Specific Pretraining and Entropy-Guided Change Detection (DSP-EntCD). First, we introduce a prior-driven training strategy called Domain-Specific Pretraining (DSP), which enhances the backbone's sensitivity to foreground information. Second, we propose an Entropy-Guided Attention Selection Mechanism (EGASM) to estimate the uncertainty of spatial locations and alleviate fusion bias between the dual-branch encoders. Furthermore, we present a Semantic-Guided Cascaded Decoder (SGCD) that integrates high-level semantics, spatial awareness, and low-level details in a complementary manner, aiming to enhance perception of multi-scale change regions and improve detection accuracy across targets of varying sizes. On the WHU-CD, LEVIR-CD, and LEVIR-CD+ datasets, which exhibit severe foreground-background imbalance, our method achieves F1 scores of 94.09%, 91.53%, and 83.98%, respectively, demonstrating SOTA detection performance.

## 1 Introduction

With the continuous advancement of remote sensing observation technology, the acquired remote sensing images are characterized by large quantity, high spatial resolution, and wide spatial coverage. As a result, change detection (CD) in remote sensing imagery has emerged as a fundamental and widely studied task in this domain, with numerous real-world applications such as urban development Buch et al. (2011), agricultural monitoring Du et al. (2022), land cover prediction Lv et al. (2022), disaster assessment Longbotham et al. (2012), and military surveillance Gong et al. (2016). To obtain more accurate change maps, deep learning-based techniques have been widely applied to change detection tasks. Methods such as DSCN Zhan et al. (2017), TransUNetCD Li et al. (2022), and ChangeMamba Chen et al. (2024) have continuously improved detection performance.

In the field of remote sensing imagery, compared to semantic segmentation tasks, change detection often faces the challenge of extreme class imbalance, as illustrated in Figure 1. Remote sensing change detection requires capturing land surface changes from bi-temporal images. However, due to the typically short intervals between image acquisitions and the inherently slow transformation of many land cover types, detectable changes are often limited—even in image pairs containing complex ground objects. Moreover, in high-resolution bi-temporal remote sensing images, change regions typically exhibit a spatial pattern of being locally concentrated yet globally sparse, which further exacerbates the spatial imbalance between foreground and background and significantly increases the difficulty of small-scale change detection.

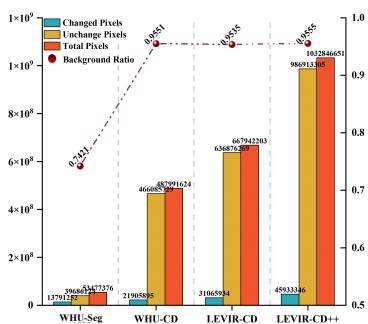

Figure 1: The core challenge in remote sensing change detection: the overwhelming dominance of background regions leads to severe class imbalance, making it difficult to identify foreground changes.

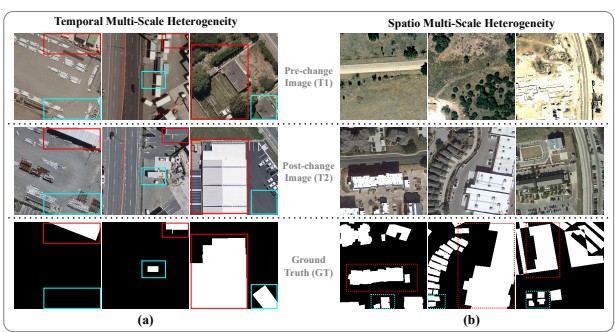

Figure 2: Visualization of the challenges posed by spatio-temporal multi-scale heterogeneity. (a) Significant scale variation of change targets within the same temporal span; (b) Uneven scale distribution of change targets within the same spatial region.

More challengingly, spatio-temporal multi-scale heterogeneity is prevalent in land cover changes: temporally, change targets exhibit differences in scale and structure within the same time span; spatially, they present scale and distribution disparities of buildings within a single image. In practical scenarios, large-scale changes tend to be coarsely detected, while small-scale changes are often overwhelmed by the background or overlooked, especially in densely built urban areas or complex terrains. These specific challenges are illustrated in Fig. 2. We propose a novel framework termed Domain-Specific Pretraining and Entropy-Guided Change Detection (DSP-EntCD), designed to address the issues of class imbalance and scale variance in change detection tasks.

To alleviate the challenges caused by the severe foreground-background imbalance, we design a two-stage training strategy called Domain-Specific Pretraining (DSP). Specifically, we first select three change detection domain-specific datasets that exclude all-background samples and design a baseline change detection model (BCD). In the first training stage, we pretrain the BCD model on the domain-specific datasets and predict the corresponding masks. Through this foreground-aware data reconstruction and domain-specific modeling strategy, BCD is guided to focus on discriminative learning in effective change regions. During the main training stage, we reuse the pretrained encoder of BCD as an Auxiliary Encoder (AE) and introduce a Main Encoder (ME) with the same architecture to form a dual-branch encoder. In addition, we design an Entropy-Guided Attention Mechanism to fuse features at corresponding scales from the dual encoders. This mechanism adaptively selects and integrates high-confidence regions based on entropy, thereby highlighting true changes in the source domain and suppressing background interference.

To address the challenge of modeling spatio-temporal multi-scale heterogeneity, we design a Semantic-Guided Cascaded Decoder (SGCD). Specifically, we incorporate the Supervised Attention Mechanism (SAM) from A2Net Li et al. (2023b) and integrate our proposed High-level Guidance Module (HGM) to obtain highly discriminative high-level semantic features, which facilitate accurate recognition of large-scale change regions across temporal sequences and spatial dimensions. In addition, to enhance the model's perception of spatio-temporal fine-grained changes, we design a Low-level Gated Guidance Module (LGGM), which dynamically filters low-level noisy features using gating weights generated from high-level semantics, thereby improving the model's sensitivity to small-scale changes such as edge details. In summary, our main contributions are as follows:

- We propose a domain-specific two-stage training strategy. Based on this strategy, we further introduce a dual-branch encoder architecture in the second stage, aiming to alleviate the extreme class imbalance in change detection tasks and enhance the model's robustness and foreground representation capability in complex scenarios.

- We propose EGASM to dynamically select features based on their information entropy, enhancing the model's response to change regions by controlling the source of features.

- We propose SGCD to achieve cascaded fusion of multi-scale features through semantic guidance. HGM incorporates high-level semantics during decoding, while LGGM lever-

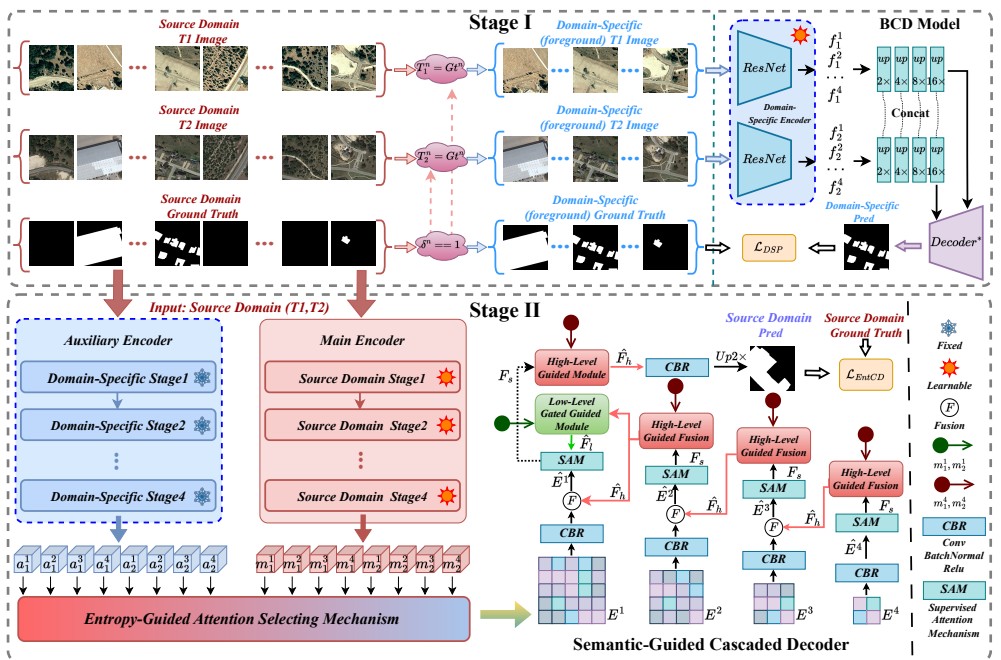

Figure 3: Overall framework of the proposed DSP-EntCD. The proposed method consists of two training stages: Stage I employs a dataset selection strategy $\delta$ to construct a domain-specific dataset, enabling the model to learn strong change-related features and provide prior knowledge of change for the encoder. Stage II introduces a dual-branch encoder architecture, where the EGASM module selects the more confident change branch, and the SGCD module performs cascaded decoding of multi-scale features to generate the final change probability map.

ages gating mechanisms to recover fine details, enabling a coarse-to-fine refined decoding process.

## 2 RLATED WORKS

In recent years, change detection (CD) has emerged as a fundamental task in remote sensing, with substantial progress achieved. Early CD methods predominantly relied on traditional pixel-based techniques Bruzzone & Prieto (2000), such as image differencing Singh (1989), image regression Ludeke et al. (1990), and principal component analysis (PCA) Richards (1984). With the rise of artificial neural networks, several early machine learning methods have been applied to CD tasks, including Support Vector Machines Bovolo et al. (2008), Decision Trees Sesnie et al. (2008), and Random Forests Seo et al. (2018). Although these methods can achieve reasonable performance in specific change scenarios, they often suffer from low accuracy and poor generalization capability. The development of remote sensing big data and artificial intelligence has brought deep learning into remote sensing image analysis. As powerful tools for feature extraction from high-resolution remote sensing data, deep learning methods such as CNNs Vinholi et al. (2022) and Huang et al. (2025), Transformers Zheng et al. (2022) and Wang et al. (2021), Mamba Zhang et al. (2025) and Wang et al. (2025), and Diffusion models Wen et al. (2024) and Zang et al. (2025) have provided new solutions for change detection.

In real-world remote sensing change detection scenarios, there exists an extreme imbalance between foreground and background regions. In recent years, many methods have been proposed to address these challenges. PCA-Net Wang et al. (2019) designed a supervised principal component analysis network combined with morphological supervised learning, leveraging pixel information near class boundaries to guide network training. AERNet Zhang et al. (2023) introduced an attention-guided edge refinement network that uses a global context aggregation module to fuse multi-level features, and a coordinate-based attention-guided decoder to capture channel and spatial dependen-

cies among features. Although the above methods have explored the class imbalance problem in change detection, they often overlook the significant scale variation of change targets. ICIF-Net Feng et al. (2022) explicitly explored the potential of integrating CNN and Transformer features through a linearized convolutional attention module, enabling bidirectional information flow at the same spatial resolution and allowing each branch to perceive the representation of the other while preserving its own characteristics. USSFC-Net Lei et al. (2023) proposed a Multi-Scale Decoupled Convolution (MSDConv) that flexibly captures multi-scale change features through recurrent multi-scale convolutions. While these methods demonstrate effectiveness in addressing specific issues, they still face limitations in more extreme environments. To address the aforementioned challenges, we propose a novel change detection network named DSP-EntCD. DSP-EntCD effectively tackles the challenges of class imbalance resulting from foreground sparsity and the modeling complexities introduced by spatio-temporal multi-scale heterogeneity, through a dual-stage training strategy and the coordinated integration of specialized modules.

## 3 METHODOLOGY

### 3.1 NETWORK ARCHITECTURE

Considering that existing end-to-end CD methods struggle with the challenges of extreme foreground-background imbalance and spatiotemporal multi-scale modeling, we propose a two-stage framework named DSP-EntCD. As shown in Figure 3, in the first stage, We design a domain-specific selection strategy $\delta$ that filters the source domain to retain only samples containing change features. The resulting domain-specific dataset consists of bi-temporal image pairs and labels that are not entirely composed of unchanged pixels. In addition, we propose a base CD model (BCD), with a $ResNet_{18}$ He et al. (2016)encoder and a simplified decoder. Notably, BCD enables the Encoder to learn strong change-related features from high change-ratio data while ensuring feature space consistency during subsequent source domain training. In the second stage, BCD's Encoder trained on the domain-specific dataset is frozen and serves as an Auxiliary Encoder (AE) in the source domain to provide stable prior change features. These features guide the Main Encoder (ME) trained on the source domain to enhance its sensitivity to change regions. The source domain training consists of four steps: 1) The frozen AE and the randomly initialized ME extract multi-scale dual-branch features from the source domain dataset. 2)An Entropy-Guided Attention Selection Mechanism (EGASM) evaluates inter-domain differences at the same feature scale and selects the more confident branch features as the main features. 3) The Semantic-Guided Cascaded Decoder (SGCD) receives multi-scale hierarchical features provided by EGASM and performs stage-wise fusion decoding of change instances. 4)The features $(m_1^4, m_2^4)$ extracted from the ME encoder are fed into the HGM to obtain coarse-grained features with strong semantic representation capabilities, which are integrated into the multi-scale decoding process. 5) In the final decoding stage, the low-level features $(m_1^1, m_2^1)$ are introduced and sent to the LGGM module to extract fine-grained features rich in spatial details.

### 3.2 OVERCOMING FOREGROUND-BACKGROUND IMBALANCE

**Domain-Specific Pretraining** Most existing remote sensing change detection datasets suffer from sparse foreground targets and an overwhelming proportion of background regions. To address this issue, we propose a Domain-Specific Pretraining (DSP) strategy, where the domain-specific refers to a subdomain dataset derived from the source domain that contains only change samples. By learning from regions that are not entirely background, we aim to enhance the encoder's ability in BCD to better recognize change regions. Specifically, we construct a foreground selection strategy based on the change labels $G_t$ from the source dataset $\mathcal{D}_{ori}$, and filter out a domain-specific training subset $\mathcal{D}_{dsp}$ with a higher proportion of changed pixels. The selection process is defined as follows:

$$\mathcal{D}_{\text{ori}} = \{(T_1^n, T_2^n), G_t^n\}_{n=1}^N, \tag{1}$$

$$\delta^n = \begin{cases} 1, & \text{if } \sum_{h=1}^{H} \sum_{w=1}^{W} G_t^n(h, w) > 0, \\ 0, & \text{otherwise,} \end{cases} \tag{2}$$

$$\mathcal{D}_{\text{dsp}} = \left\{ [(T_1^n, T_2^n), Gt^n] \in \mathcal{D}_{\text{ori}} \mid \delta^n = 1 \right\}, \tag{3}$$

where $N$ denotes the total number of samples in $\mathcal{D}_{ori}$, $H$ and $W$ represent the height and width of $G_t$ respectively, and $\delta_n$ indicates whether the $n$ sample contains changed pixels. If $\delta_n = 1$, the sample is included in $\mathcal{D}_{dsp}$.

On the domain-specific dataset, we use $ResNet_{18}$ as the encoder for BCD and design a simple convolutional decoder to prevent downstream modules from interfering with the learning of change features. In addition, to enhance the encoder's ability to discriminate change regions and improve boundary sensitivity, we introduce a joint loss function composed of BCE and Dice loss during training, which is consistent with the loss used in the second stage. The training loss is defined as:

$$\mathcal{L}_{\text{dsp}} = \mathcal{L}_{\text{BCE}}(Y_{dsp}, \hat{Y_{dsp}}) + \mathcal{L}_{\text{Dice}}(Y_{dsp}, \hat{Y_{dsp}}), \tag{4}$$

where $Y_{\text{dsp}}$ denotes the ground truth labels from the domain-specific dataset, and $\hat{Y}_{\text{dsp}}$ represents the corresponding model predictions.

**Knowledge-Frozen Dual-Stream Encoder**    In the second stage of DSP-EntCD, we design a dual-branch parallel encoder based on the DSP strategy. Specifically, the $ResNet_{18}$ model trained on the domain-specific dataset is frozen during the second stage and used as the Auxiliary Encoder (AE) to provide stable change priors. Meanwhile, another randomly initialized $ResNet_{18}$ is employed as the Main Encoder (ME) to progressively learn the feature distribution and change patterns of the source domain dataset. Given an input bi-temporal remote sensing image pair $(T1, T2) \in D_{ori}$, the feature extraction processes of AE and ME are denoted as $\mathcal{F}_a(\cdot)$ and $\mathcal{F}_m(\cdot)$, respectively. The final outputs of the encoder are:

$$F_a = \mathcal{F}_a(T_1, T_2), F_m = \mathcal{F}_m(T_1, T_2), \tag{5}$$

Where the specific feature outputs of AE and ME are $F_a = \{a_1^i, a_2^i \mid i = 1, 2, 3, 4\}$ and $F_m = \{m_1^i, m_2^i \mid i = 1, 2, 3, 4\}$, respectively.

**Entropy-Guided Attention Selection Mechanism**    Information entropy is an important statistical measure used to quantify the uncertainty of random variables. It can be employed to assess the confidence of heterogeneous feature maps at each spatial location. Therefore, we use entropy as a guiding signal to dynamically adjust the fusion weights of multi-source feature maps extracted by the dual-branch encoders, enabling the model to select more reliable change features while suppressing background noise.

Specifically, the Entropy-Guided Attention Selection Mechanism (EGASM) consists of two key components: entropy-aware feature alignment and entropy-based strategic selection. The architecture of the module is provided in the Supplementary Material. On the one hand, to enhance the stability and discriminability of entropy computation, we introduce an attention-guided fusion module before entropy is applied. This module integrates the original bi-temporal feature maps extracted by the source and domain-specific encoders through guided alignment. Concretely, we concatenate the temporally heterogeneous but source-consistent features $(m_1^i, m_2^i)$ and $(a_1^i, a_2^i)$ to obtain preliminary change feature maps $m^i$ and $a^i$, respectively. Then, global average pooling and $Conv_{7 \times 7}$ are used to extract prominent spatial features $m_s^i$, $a_s^i$ and channel-wise features $m_c^i$, $a_c^i$. Finally, attention weights $w_m^i, w_a^i$ are generated through Hadamard Product $\odot$ of these two types of features:

$$w_m^i = m_s^i \odot m_c^i, w_a^i = a_s^i \odot a_c^i. \tag{6}$$

Finally, the attention weights are used to perform weighted fusion of $(m_1^i, m_2^i)$ and $(a_1^i, a_2^i)$, providing stable and semantically enhanced inputs for the subsequent entropy-based selection:

$$m_w^i = m_1^i \cdot w_m^i + m_2^i \cdot (1 - w_m^i), \tag{7}$$

$$a_w^i = a_1^i \cdot w_a^i + a_2^i \cdot (1 - w_a^i), \tag{8}$$

where $m_w^i$ and $a_w^i$ denote the stable change features of ME and AE at the $i$-th scale, respectively.

On the other hand, although the initial attention-based fusion aligns and enhances homologous features, the ME and AE branches still exhibit distinct inductive biases. Simply concatenating $m_w^i$ and $a_w^i$ is insufficient to obtain accurate change features. To address this, we introduce an entropy-based mechanism that compares their channel-wise entropy and selects the more confident features as the

dominant input. This enables more refined and robust fusion. Specifically, we apply softmax along the channel dimension to obtain normalized probability distributions:

$$\hat{m_w^i} = \text{softmax}(m_w^i), \hat{a_w^i} = \text{softmax}(a_w^i). \tag{9}$$

Based on the definition of information entropy, calculate the channel-wise entropy of the two branch features $\hat{m_w^i}$ and $\hat{a_w^i}$:

$$e_m^i = -\sum_c \hat{m_w^i} \cdot \log(\hat{m_w^i} + \epsilon), e_a^i = -\sum_c \hat{a_w^i} \cdot \log(\hat{a_w^i} + \epsilon), \tag{10}$$

where $\epsilon$ is a stabilizing term to prevent $\log(0)$. Finally, the channel-wise entropy difference is computed as the fusion weight, reflecting the spatial uncertainty of ME features, while AE provides prior guidance. The process is defined as:

$$\Delta e = \sigma \left( \beta \cdot (e_m^i - e_a^i) \right), \tag{11}$$

$$E^i = (1 - \Delta e) \cdot m_w^i + \Delta e \cdot a_w^i, \tag{12}$$

where $\Delta e$ denotes the entropy difference, $\sigma$ represents the Sigmoid function, and $\beta$ is a scaling factor that controls the sensitivity of $\sigma$ to the entropy difference. $E^i$ represents the weighted fused features of the dual-branch change maps, which are used for subsequent decoding of the change map.

### 3.3 SPATIO-TEMPORAL MULTI-SCALE HETEROGENEITY

**Semantic-Guided Cascaded Decoder** As shown in Figure 2, to address the challenge of spatio-temporal multi-scale heterogeneity in change detection, we design a cascaded architecture with progressive decoding, where a SAM is integrated into each decoding stage to establish a foreground/background saliency-guided mechanism. Specifically, SAM first applies $Conv_{1\times1}$ to the input feature $\hat{E}$ to generate a foreground probability map $P_f$, and derives the corresponding background map as $P_b$. The two maps are then concatenated along the channel dimension to form a foreground-background saliency map. This map is passed through a $1 \times 1 \, CBR$ block and element-wise multiplied with the original feature $\hat{E}$. Finally, the result is fed into a $3 \times 3 \, CBR$ block to further extract saliency-enhanced features $F_{SAM}$, thereby achieving guidance enhancement at the current scale. The process can be formulated as follows:

$$P_f = \sigma(Conv_{1\times1}(\hat{E})), \quad P_b = 1 - P_f, \tag{13}$$

$$F_{SAM} = CBR^3(\hat{E} \odot CBR^1(Cat(P_f, P_b))), \tag{14}$$

Where $Cat$ denotes channel-wise concatenation of features, $CBR$ represents a sequence of Conv, BatchNorm, and ReLU layers.

In remote sensing tasks, high-level semantic features are relatively stable and help maintain semantic consistency of change across stages during the coarse-to-fine cascade decoding process. However, the top-level outputs extracted by the original encoder are still a pair of features from two temporal inputs, without explicit temporal modeling, and therefore cannot be directly used for semantic guidance in decoding. To address this, we construct a multi-scale semantic-guided cascade decoder. Specifically, we propose the HGM, where the bi-temporal features $m_1^i$ and $m_2^i$ from the source encoder are first concatenated and then enhanced via $1 \times 1 \, CBR$ block to obtain the high-level temporal semantic feature $F_h$. Finally, we employ multiple $CBR$ blocks to align the channels of $F_h$ and the current-scale feature $F_{SAM}$, and fuse them. This process can be formulated as:

$$F_h = CBR^1(Cat(m_1^4, m_2^4)), \tag{15}$$

$$\hat{F}_h = CBR \left( Cat(CBR(F_{SAM}, F_h^{\uparrow})) \right), \tag{16}$$

where $\uparrow$ denotes feature upsampling, and $\hat{F}_h$ represents the final output of the HGM, and $h$ indicates the explicit temporally-aware features.

Finally, at each decoding stage, we fuse the current-stage features with the upsampled features from the previous stage. By introducing high-level semantic signals from a consistent source into each stage, the model maintains semantic coherence in its decision-making process, reduces error propagation caused by scale misalignment or low-level noise, and effectively improves the detection accuracy of change targets across different scales as well as the reconstruction of their boundaries.

Table 1: Compare our model with other methods on WHU-CD, LEVIR-CD and LEVIR-CD+ datasets. $^*$ indicates that the encoder was trained based on $ResNet_{18}$ weights pre-trained on ImageNet. Bold indicates the best performance.

| Methods | WHU-CD | | | | LEVIR-CD | | | | LEVIR-CD+ | | | |
|---|---|---|---|---|---|---|---|---|---|---|---|---|
| | Pre. | Rec. | F1. | IoU | Pre. | Rec. | F1. | IoU | Pre. | Rec. | F1. | IoU |
| FC-EF | 79.23 | 73.45 | 76.23 | 61.59 | 75.90 | 61.12 | 67.71 | 51.19 | 59.49 | 51.01 | 54.92 | 37.86 |
| BIT | 94.57 | 88.86 | 91.63 | 84.55 | 90.68 | 90.15 | 90.42 | 82.51 | 82.42 | 79.54 | 80.95 | 67.99 |
| A2Net | 92.23 | 90.48 | 91.35 | 84.07 | 90.13 | 89.21 | 89.67 | 81.27 | 78.67 | 78.61 | 78.64 | 64.80 |
| SEIFNet | 93.45 | 89.68 | 91.53 | 84.38 | 91.14 | 89.16 | 90.14 | 82.05 | 81.24 | 79.23 | 80.22 | 66.97 |
| BiFA | 93.74 | 91.64 | 92.68 | 86.36 | 92.26 | 90.29 | 91.27 | 83.94 | 81.56 | 82.29 | 81.92 | 69.38 |
| ChangeRD | 91.81 | 90.31 | 91.05 | 83.57 | 91.03 | 89.34 | 90.18 | 82.12 | 82.21 | 79.67 | 80.92 | 67.95 |
| ISDANet | 92.18 | 93.88 | 93.03 | 86.96 | 90.17 | 90.34 | 90.26 | 82.24 | 73.19 | 87.45 | 79.69 | 66.24 |
| STRobustNet | 94.22 | 92.89 | 92.55 | 87.88 | 92.16 | 90.40 | 91.27 | 83.95 | 83.58 | 81.92 | 82.75 | 70.57 |
| DSP-EntCD | 95.22 | 90.87 | 92.99 | 86.90 | 92.57 | 90.52 | 91.53 | 84.39 | 86.90 | 80.20 | 83.41 | 71.54 |
| DSP-EntCD* | 95.10 | 93.10 | 94.09 | 88.83 | 91.65 | 90.55 | 91.10 | 83.65 | 85.10 | 82.89 | 83.98 | 72.38 |

**Fine-Grained Structural Information**    Although the cascaded decoder guided by high-level semantics is effective in capturing hierarchical contextual information, it still lacks sensitivity to fine-grained structures such as object boundaries and subtle textures. To address this issue, we propose the LGGM. It is worth noting that shallow features are introduced only at the final decoding stage to enhance the model's representation of small-scale changes. This design is mainly based on the following considerations: on one hand, injecting low-level features too early may interfere with semantic guidance; on the other hand, detailed information becomes semantically meaningful only when the decoding has restored high-resolution spatial representations.

To alleviate the temporal heterogeneity in low-level features, we first concatenate and fuse the shallow features $m_1^1$ and $m_2^1$ from the ME branch along the channel dimension. High-level semantic features are then introduced as the gating source. This process can be formulated as:

$$F_l = CBR^1(m_1^1, m_2^1), \tag{17}$$

$$G = \sigma(CBR^1(F_h)) \odot F_l, \tag{18}$$

Where $F_l$ denotes the low-level detail features with temporal information, $F_h$ represents the high-level semantic temporal features, and $G$ refers to the gating coefficients used to filter noise in the low-level features.

Finally, the gating coefficients $G$ are used to weight the fused low-level features, generating a spatially selective mask. The filtered detail features are then fused with the high-level semantics to form the final representation $\hat{F}_l$:

$$F_G = G \odot CBR^1(m_1^1, m_2^1), \tag{19}$$

$$\hat{F}_l = F_G + F_l, \tag{20}$$

Where $\odot$ denotes element-wise multiplication, and $\hat{F}_l$ represents the denoised detail features enriched with texture and edge information.

## 4 EXPERIMENTS

### 4.1 EXPERIMENTAL SETUP

To verify the effectiveness of our proposed DSP-EntCD in the field of remote sensing change detection, we conducted experiments on three public change detection datasets characterized by multi-scale features and extremely low foreground ratios, namely WHU-CD Ji et al. (2019), LEVIR-CD, and LEVIR-CD+ Chen & Shi (2020). Detailed information about these datasets can be found in the supplementary material. All experiments were conducted on an NVIDIA GeForce RTX 3090 GPU, with input image sizes uniformly resized to 256 × 256 pixels. Simple data augmentation techniques, including normalization, vertical flipping, and horizontal flipping, were employed. The batch was set to 8, and the Adam optimizer was used.

To reduce the domain discrepancy between AE and ME, we adopted the same learning rate (1e-4) and loss functions (BCE Loss and Dice Loss) in both the first and second training stages, ensuring

Table 2: Ablation study of proposed modules. When both Stage 1 and Stage 2 are enabled, the backbone model pre-trained in Stage 1 is reused as the AE in Stage 2.

| Stage1 | Stage2 | | | WHU-CD | | LEVIR-CD | | LEVIR-CD+ | |
|---|---|---|---|---|---|---|---|---|---|
| | ME | EGASM | SGCD | F1. | IoU | F1. | IoU | F1. | IoU |
| ✓ | ✗ | ✗ | ✗ | 89.70 | 81.32 | 90.20 | 82.15 | 81.89 | 69.33 |
| ✗ | ✓ | ✗ | ✗ | 91.09 | 83.64 | 90.50 | 82.65 | 81.75 | 69.13 |
| ✓ | ✓ | ✗ | ✗ | 91.74 | 84.74 | 90.64 | 82.89 | 82.27 | 69.88 |
| ✓ | ✓ | ✓ | ✗ | 91.93 | 85.07 | 90.73 | 83.03 | 82.86 | 70.74 |
| ✓ | ✓ | ✓ | ✓ | **92.99** | **86.90** | **91.53** | **84.39** | **83.41** | **71.54** |

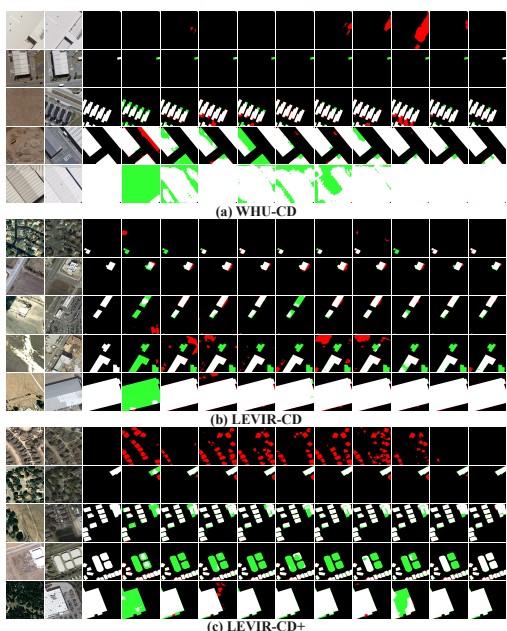

Figure 4: Visual results under different foreground-background ratios and object scales. DSP-EntCD* demonstrates lower false positives and miss rates.

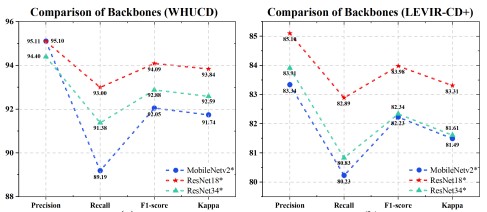

Figure 5: Ablation study of backbone. The red curve represents the quantitative performance of the baseline model finally adopted by DSP-EntCD.

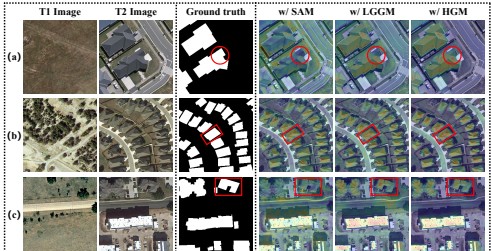

Figure 6: Ablation study on the module heatmap results of SGCD in the $E^1$ decoding stage.

consistent optimization for both modules. For evaluation, we employed the most commonly used and authoritative metrics in remote sensing change detection, including Overall Accuracy (OA.), Precision (Pre.), Recall (Rec.), F1 Score (F1.), IoU, and the Kappa coefficient.

### 4.2 COMPARISON WITH STATE-OF-THE-ART METHODS

We conducted both quantitative and qualitative comparisons between our proposed method and eight state-of-the-art (SOTA) approaches, including FC-EF Caye Daudt et al. (2018), BIT Chen et al. (2022), A2Net Li et al. (2023a), SEIFNet Huang et al. (2024), BiFA Zhang et al. (2024), ChangeRD Jing et al. (2025), ISDANet Ren et al. (2025), and STRobustNet Caye Daudt et al. (2018). Table 1 presents the performance comparison between DSP-EntCD and the aforementioned methods on three datasets characterized by extreme foreground-background imbalance. Our method achieves the highest F1 score across all datasets, clearly demonstrating its accuracy and completeness in detecting small-scale foreground changes. To intuitively demonstrate the effectiveness of DSP-EntCD in handling extremely imbalanced foreground-background ratios and spatio-temporal multi-scale scenarios, Figure 4 presents the visual comparison results of different methods on the three datasets (in each group, the degree of change and target scale increase progressively from top to bottom).

Table 3: Ablation study on the scaling factor $\beta$ controlling entropy sensitivity in EGASM.

| Entropy Scaling Factor | WHU-CD | | | LEVIR-CD | | | LEVIR-CD+ | | |
|---|---|---|---|---|---|---|---|---|---|
| | OA. | F1. | IoU. | OA. | F1. | IoU. | OA. | F1. | IoU. |
| $\beta == 0.0$ | 99.43 | 92.89 | 86.73 | 99.09 | 90.8 | 83.15 | 98.68 | 82.35 | 69.99 |
| $\beta == 3.0$ | 99.53 | 94.06 | 88.79 | 99.1 | 91.08 | 83.61 | **98.73** | **84.11** | **72.57** |
| $\beta == 5.0$ | **99.54** | **94.09** | **88.83** | **99.1** | **91.09** | **83.64** | 98.71 | 83.98 | 72.38 |
| $\beta == 8.0$ | 99.54 | 94.08 | 88.81 | 99.09 | 91.09 | 83.64 | 98.69 | 83.80 | 72.12 |

## 4.3 ABLATION STUDY

**Proposed Modules**  To further validate the effectiveness and robustness of our proposed two-stage training strategy and its components, we conducted ablation studies on three datasets: WHU-CD, LEVIR-CD, and LEVIR-CD+. Based on the DSP-EntCD model architecture, we designed five ablation settings:1) Training the baseline model on the domain-specific dataset and evaluating on the source dataset; 2) Training the baseline model directly on the source dataset and predicting on the same; 3) Freezing the baseline model weights trained on the domain-specific dataset, using its encoder as the Auxiliary Encoder (AE) in Stage 2, combined with the Main Encoder (ME) to form a dual-branch encoder for source dataset prediction; 4) Introducing the EGASM module on top of setting 3 to select more confident features for decoding; 5) Incorporating the SGCD module to enhance multi-scale feature decoding capability. As shown in Table 2, the experimental results are presented in sequence according to the above settings. It can be observed that with the improvement of the training strategy and the introduction of the proposed modules, the F1 scores steadily increase across all three datasets, demonstrating that our model can better handle extreme foreground-background imbalance and multi-scale land cover challenges. In addition, we further evaluated the impact of different encoder backbones on the model's performance. As shown in Figure 5, we compare the overall performance of DSP-EntCD with different backbones, including MobileNetV2Sandler et al. (2018), ResNet18, and ResNet34. The results show that ResNet18 achieves the best trade-off between accuracy and efficiency, delivering the highest overall performance.

**Entropy Scaling Factor And Semantic-Guided Cascaded Decoder**  As described in Section 3.2.3, the parameter $\beta$ is used to control the sharpness of the entropy difference weight distribution, thereby regulating the response to entropy differences during feature fusion. Table 3 shows that the model achieves the best performance when $\beta = 5$, where the entropy difference is moderately amplified, resulting in stable and effective fusion. A too small $\beta$ causes the fusion to degrade into a linear average with insufficient information selection, while a too large $\beta$ leads to most positions retaining only a single branch, causing feature loss.

To evaluate the contribution of the SGCD module to the final decoding performance of DSP-EntCD, we conducted ablation experiments on the proposed LGGM and HGM modules. As shown in Figure 6, we selected three representative pairs of bi-temporal remote sensing images along with their corresponding labels. The attention heatmaps of each module are overlaid on the T2 image for visualization. In the heatmaps, brighter colors indicate stronger attention responses, while darker colors suggest lower attention. Overall, the proposed modules help the model better decode multi-scale features.

## 5 CONCLUSION

We propose DSP-EntCD to address the prevalent challenges of extreme foreground-background imbalance and spatiotemporal multi-scale heterogeneity in remote sensing imagery. Specifically, to tackle the issue of the extremely small proportion of change regions, we design a dual-stage training strategy that leverages a domain-specific pretrained BCD as an AE in the second stage to provide change priors for the ME. In addition, we introduce the EGASM to guide feature fusion between the dual encoders and enhance critical features, enabling the model to better distinguish between foreground and background. To address the challenges of multi-scale changes, we incorporate the SGCD to achieve precise localization of global, local, and spatially multi-scale entities within the image. Extensive experimental results validate the effectiveness of DSP-EntCD in change detection tasks.

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
