# OpenReview forum: "DSP-EntCD: A Knowledge-Freezing, Entropy-Guided Remote Sensing Change Detection Network with Domain-Specific Pretraining"
_ICLR.cc/2026/Conference — ICLR 2026 Conference Withdrawn Submission_

### Official Review · Reviewer_oDHG · 2025-10-19

**Soundness:** 2
**Presentation:** 2
**Contribution:** 1
**Rating:** 2
**Confidence:** 4

**Summary:**

The paper proposes DSP-EntCD, a two-stage knowledge-freezing framework for remote sensing change detection. Stage I performs Domain-Specific Pretraining (DSP) by filtering out background-only tiles to pretrain a baseline CD encoder; Stage II freezes this encoder as an Auxiliary Encoder (AE), pairs it with a learnable Main Encoder (ME), and fuses them via an Entropy-Guided Attention Selection Mechanism (EGASM). Decoding uses a Semantic-Guided Cascaded Decoder (SGCD) with a high-level guidance module (HGM) and a low-level gated guidance module (LGGM). The method reports SOTA F1 on WHU-CD, LEVIR-CD, and LEVIR-CD+.

**Strengths:**

- A clear, end-to-end recipe (DSP + EGASM + SGCD) with a clean system design.
- Method is formula-tized (e.g., entropy fusion, SAM) with a helpful overview figure.

**Weaknesses:**

This manuscript contains insufficient engagement with closely related work.

1. The core change-aware pretraining motivation, biasing representation toward change via objectives/data that emphasize temporal variation, parallels prior change-aware sampling / contrastive and seasonality-aware pretraining (e.g., Mall et al., CVPR’23; Mañas et al., ICCV’21). The paper does not cite or experimentally compare against these families, so it is unclear whether DSP is preferable to established change-aware pretraining on similar data budgets.
  - Mall, Utkarsh, Bharath Hariharan, and Kavita Bala. "Change-aware sampling and contrastive learning for satellite images." Proceedings of the IEEE/CVF Conference on Computer Vision and Pattern Recognition. 2023.
  - Manas, Oscar, et al. "Seasonal contrast: Unsupervised pre-training from uncurated remote sensing data." Proceedings of the IEEE/CVF International Conference on Computer Vision. 2021.


2. “Knowledge-frozen dual-stream” is conceptually close to side-tuning/adapters.
Freezing a pretrained backbone and learning a parallel stream that is fused during decoding is very reminiscent of Side-Tuning (Zhang et al., ECCV’20). The manuscript credits “knowledge-freezing two-stage training” as a contribution, but the adaptation pattern and motivation appear derivative; there is no baseline that re-implements side-tuning or a modern adapter to establish novelty/benefit over that paradigm.
  - Zhang, Jeffrey O., et al. "Side-tuning: a baseline for network adaptation via additive side networks." European conference on computer vision. Cham: Springer International Publishing, 2020.

3. SGCD is not orthogonal to cascaded/semantic-guided decoders.
The cascaded, coarse-to-fine semantic guidance resembles earlier cascaded dual-decoders and supervised-attention guided decoding for CD and dense prediction (e.g., Yang et al., TGRS’24; Li et al., TGRS’23; Zhang et al., PR’25). Without direct, controlled comparisons (same backbone/training), the manuscript does not establish that SGCD is meaningfully new or better than these semantically guided cascades.
  - Yang, Feng, et al. "Change-aware cascaded dual-decoder network for remote sensing image change detection." IEEE Transactions on Geoscience and Remote Sensing 62 (2024): 1-12.
  - Li, Zhenglai, et al. "Lightweight remote sensing change detection with progressive feature aggregation and supervised attention." IEEE Transactions on Geoscience and Remote Sensing 61 (2023): 1-12.
  - Zhang, Gang, et al. "CEDNet: A cascade encoder–decoder network for dense prediction." Pattern Recognition 158 (2025): 111072

4. Uncertainty-guided attention is under-cited and under-compared.
EGASM relies on information entropy to guide fusion, which is a specific instance of uncertainty-guided attention/selection. There is a substantial body of work using uncertainty to gate attention/features in vision and medical/RS tasks (e.g., He et al., 2023; Xiong et al., 2025). None are cited or compared, leaving the contribution positioning ambiguous.
  - He, Wei, et al. "Building extraction from remote sensing images via an uncertainty-aware network."Transactions on Geoscience and Remote Sensing. 2024.
  - Xiong, Hao, et al. "Uncertainty-guided attention learning for malaria parasite detection in thick blood smears." Neural Networks 191 (2025): 107833.

**Questions:**

- Positioning vs. change-aware pretraining:
How does DSP compare to Mall et al. (CVPR’23) and Mañas et al. (ICCV’21) under equal backbones and data? Please run controlled pretraining ablations (same epochs, same unlabeled/labeled budget) and report CD metrics.
- Cascaded decoder comparisons:
Please compare SGCD against (a) Yang et al., TGRS’24 dual-decoder cascade and (b) Li et al., TGRS’23 progressive aggregation + supervised attention, using the same encoder and training recipe.
- Uncertainty-guided attention ablations:
Beyond entropy, evaluate (i) variance-based Bayesian channel attention, (ii) Monte-Carlo dropout entropy, and (iii) temperature-scaled confidence maps as the gating signal, with calibration plots. Report whether β interacts with calibration.

---

### Official Review · Reviewer_whfR · 2025-10-23

**Soundness:** 3
**Presentation:** 2
**Contribution:** 3
**Rating:** 4
**Confidence:** 3

**Summary:**

- This paper addresses two core challenges in remote sensing change detection: extreme foreground-background imbalance and spatiotemporal multi-scale heterogeneity. To solve these issues, the authors propose a two-stage framework called DSP-EntCD, which integrates three key components, i.e. Domain-Specific Pretraining (DSP), Entropy-Guided Attention Selection Mechanism (EGASM), and Semantic-Guided Cascaded Decoder (SGCD).
- For validation, the authors conducted experiments on three public datasets (WHU-CD, LEVIR-CD, LEVIR-CD+) with extreme class imbalance, using metrics including F1 Score, IoU, and OA. The results show DSP-EntCD achieves SOTA performance (F1 scores of 94.09%, 91.53%, 83.98% on the three datasets, respectively).
- Ablation studies further verify the effectiveness of the two-stage strategy and key modules, and ResNet18 is confirmed as the optimal backbone balancing accuracy and efficiency.

**Strengths:**

> * 1. The two-stage DSP strategy mitigates foreground-background imbalance by pre-training on a background-free domain-specific dataset，avoiding the limitation of traditional single-stage training where models are biased toward background. Meanwhile, EGASM and SGCD complement each other—EGASM selects high-confidence features via entropy to reduce fusion bias, while SGCD achieves coarse-to-fine multi-scale feature fusion through HGM and LGGM. This design addresses the two points of remote sensing CD.
> - 2. The paper are well-writen. the Experiments section presents results systematically with tables and figures; the Conclusion section concisely summarizes contributions. Symbols and technical terms are used accurately, enhancing readability and academic rigor.
> - 3. Ablation studies further verify the effectiveness of the proposed framework.

**Weaknesses:**

>- 1. The paper does not test its performance in extreme complex scenarios common in remote sensing, such as: 1) Scenes with large seasonal variations; 2) Scenes with small-scale subtle changes (e.g., road crack expansion). This limits the generalization of its conclusions. If possible, it is better to show some cases in these extreme complex scenarios.
>- 2. The paper focuses on accuracy metrics (F1 Score, IoU, OA) when comparing with SOTA methods and validating modules, but does not report key computation cost indicators such as model parameter count (Params), floating-point operations (FLOPs), inference time per image, or training time. For remote sensing CD, which often processes large-scale high-resolution images, computation efficiency is critical for practical deployment. For example, if the proposed method achieves higher F1 Score than SOTA but has 2x more FLOPs or 3x longer inference time, its applicability in real-time or resource-constrained scenarios would be limited. This omission makes it impossible to evaluate the method’s trade-off between accuracy and efficiency comprehensively.
>- 3. In Section 4.3, the authors evaluate HGM and LGGM (sub-modules of SGCD) through attention heatmap visualization (Figure 6)，claiming “the proposed modules help the model better decode multi-scale features” based on “brighter colors indicating stronger attention responses”. However, visual evaluation is subjective (e.g., different readers may judge “attention alignment with change regions” differently) and lacks objective quantitative support. This makes the conclusion about SGCD’s contribution less rigorous.

**Questions:**

>- 1. Section 3.3 mentions that “shallow features are introduced only at the final decoding stage” to avoid interfering with semantic guidance and ensure detailed information is semantically meaningful. However, this claim is not supported by comparative experiments—there is no ablation study testing the impact of introducing LGGM in the 2nd or 3rd decoding stage on small-scale change detection accuracy. Would you please provide experimental evidence or more detailed theoretical explanations to justify this design choice.
>- 2. The paper states that the AE (frozen BCD encoder) provides “stable change priors” to guide the ME，but does not explain: 1) How the ME absorbs the AE’s prior knowledge (e.g., through feature similarity constraints or loss function guidance); 2) Whether the frozen AE causes the ME to fall into "local optima" (e.g., over-relying on the AE’s features and ignoring new patterns in the source domain). Please supplement an analysis of the knowledge transfer mechanism between AE and ME，and if possible, add experimental results to verify the effectiveness of “knowledge freezing”.

---

### Official Review · Reviewer_16xf · 2025-11-01

**Soundness:** 3
**Presentation:** 2
**Contribution:** 2
**Rating:** 4
**Confidence:** 5

**Summary:**

This paper proposes DSP-EntCD, a two-stage framework for remote sensing change detection that integrates both stages into a single pipeline: in Stage 1, Domain-Specific Pretraining (DSP) pre-trains a baseline CD encoder using only change-containing samples, and in Stage 2, the pre-trained encoder is frozen as an auxiliary encoder (AE) and paired with a trainable main encoder (ME) to form a dual-branch architecture. Within this architecture, an Entropy-Guided Attention Selection Mechanism (EGASM) adaptively selects and fuses multi-scale features according to entropy confidence, and a Semantic-Guided Cascaded Decoder (SGCD) comprising a High-level Semantic Guidance Module (HGM) and a Low-level Gated Guidance Module (LGGM) jointly captures large-scale and fine-grained changes.

**Strengths:**

Pretrain on a subdomain containing only change samples, then apply knowledge freezing to supply a prior to the main branch, directly addressing the severe foreground and background imbalance in change detection. The decoder uses a coarse-to-fine design that fits the multi-scale and boundary requirements of change detection.

**Weaknesses:**

1. The Entropy-Guided Attention Selection Mechanism is a key step for assessing the confidence of the AE and ME, but the experimental analysis does not intuitively show how it works; the correspondence between heatmaps and entropy maps should be analyzed.
2. High-resolution inference for remote sensing is cost-sensitive. The computational cost and energy consumption are not clearly explained. Authors only mention training on a single RTX 3090 GPU and do not report per-stage time, Flops, GPU memory usage, or throughput.
3. The problem of a “stale prior” caused by freezing the AE is not discussed, nor whether periodic updates to the AE are needed to mitigate “knowledge aging.” The issue of data generalization is also not addressed; for example, the AE is pre-trained only on LEVIR-CD, then Stage 2 evaluates change detection performance on WHU-CD.

**Questions:**

1. Is the DSP subdomain defined solely from the training set, and which datasets does it come from? Were the validation and test sets completely excluded from the selection? Please report the proportion of samples included in the subdomain.
2. The ablation granularity is insufficient. Table 2 presents incremental, module-by-module additions, but lacks a full-factor ablation to characterize module interactions, such as whether EGASM and SGCD are complementary or redundant. The individual ablations of HGM and LGGM are also not presented.

---

### Official Review · Reviewer_tJd1 · 2025-11-01

**Soundness:** 2
**Presentation:** 2
**Contribution:** 3
**Rating:** 4
**Confidence:** 3

**Summary:**

This paper addresses the extreme foreground-background imbalance and spatiotemporal multi-scale heterogeneity problems in remote sensing change detection by proposing the DSP-EntCD framework, which operates through three core innovative components working synergistically: First, it adopts a Domain-Specific Pretraining (DSP) strategy, where an auxiliary encoder is pretrained on a subset containing only changed samples and then frozen, effectively alleviating class imbalance; Second, it designs an Entropy-Guided Attention Selection Mechanism (EGASM) that utilizes information entropy to quantify feature uncertainty and dynamically fuses features from dual-branch encoders; Finally, it constructs a Semantically-Guided Cascaded Decoder (SGCD) that achieves multi-scale feature fusion through HGM and LGGM modules. This method achieves F1 scores of 94.09%, 91.53%, and 83.98% on the WHU-CD, LEVIR-CD, and LEVIR-CD+ datasets respectively, surpassing existing SOTA methods. The main advantages of this paper include: the innovative combination of domain-specific pretraining with a knowledge freezing mechanism, a theoretically sound entropy-guided feature selection strategy, comprehensive ablation experiments and visualization validation, and practical value in extreme imbalance scenarios, providing an effective solution for the field of remote sensing change detection.

**Strengths:**

Innovative Two-Stage Training Strategy: Domain-specific pretraining on change-only samples innovatively alleviates extreme class imbalance; the knowledge freezing mechanism uses the pretrained encoder as an auxiliary branch, ensuring feature space consistency and preventing catastrophic forgetting.

Entropy-Guided Feature Selection: Information entropy quantifies feature uncertainty, enabling adaptive fusion of dual-branch encoder features through dynamic attention weighting; this theoretically sound mechanism intelligently selects reliable representations based on feature quality, surpassing simple concatenation or averaging.

Comprehensive Experimental Validation :Thorough evaluation on WHU-CD, LEVIR-CD, and LEVIR-CD+ benchmarks with detailed ablation studies verifying each module's effectiveness; rich visualizations demonstrate detection capabilities across diverse scenarios including building changes and urban expansion.

Significant Practical Value: Addresses core challenges of extreme foreground-background imbalance and multi-scale detection in real-world remote sensing; achieves 83.98% F1 on the extremely imbalanced LEVIR-CD+ dataset, demonstrating strong robustness and generalization.

**Weaknesses:**

Crude Dataset Construction Strategy The strategy only excludes full-background samples (Σ Gt(h,w) > 0) without considering changed pixel ratio, treating 1-pixel changes equal to 50% changes. This leaves "quasi-background" samples (<0.1% change) in Ddsp, causing domain shift and overfitting. Recommend threshold-based filtering: δ^n = 1 if (Σ Gt^n(h,w))/(H×W) > τ, with ablation on τ ∈ [0.01, 0.05, 0.10] and reporting Ddsp statistics.

Weak Theoretical Foundation for EGASM Using entropy difference as confidence lacks justification—low entropy may indicate limited expression, not high quality. Hyperparameter β (3.0-5.0, 1% F1 gain) is sensitive with unclear selection basis; ε undefined risks gradient vanishing. Recommend theoretical explanation linking entropy to feature quality, visualize β search process, and compare with variance/Gini coefficient.

Incomplete Comparisons and Questionable SOTA Claims Table 1 lacks recent Transformer/Mamba methods (Swin-CD, ChangeMamba, CDMamba); STRobustNet's IoU (87.88%) exceeds this work (86.90%) on WHU-CD despite "SOTA" claims based on F1; training settings unclear. Recommend adding 2024-2025 methods, reporting both F1 and IoU, clarifying training fairness, and honestly discussing relative strengths.

**Questions:**

Q1: Ddsp Statistical Characteristics Need exact Ddsp proportions in WHU-CD, LEVIR-CD, LEVIR-CD+ (if too high, what's the point of "domain-specific"?), average changed pixel ratio with distribution histogram, and justification for choosing τ=0 over stricter thresholds (τ=0.05, 0.10). These clarify DSP effectiveness.

Q2: Auxiliary Encoder in Inference Clarify if AE participates during inference and its speed impact. Table 2 shows AE adds only 0.65% F1 (91.09%→91.74%)—does this justify dual-branch complexity? Critically missing: single-branch ME-only baseline trained from scratch to prove dual-branch necessity.

Q3: β Learnability Table 3 shows optimal β=5.0 for WHU-CD and LEVIR-CD+, but LEVIR-CD missing. Is β consistent across datasets? Explore making β learnable (e.g., MLP predicting β dynamically) to avoid manual tuning, and test alternative functions beyond scaling+Sigmoid (Softmax, Tanh) for robustness.

Q4: Relationship with ImageNet Pretraining Table 1 shows ImageNet pretraining hurts WHU-CD (94.09%→91.10%, -3%) but helps LEVIR-CD+ (83.41%→83.98%). Need experiments combining ImageNet+DSP (ImageNet init→Ddsp fine-tune) to understand DSP's role, and explain this inconsistency.

Q5: HGM-LGGM Coupling Equation (18) uses Stage4 Fh to gate Stage1 features despite semantic gap—poor HGM harms LGGM. Test independent gating networks for decoupling, visualize G to verify edge activation, and ablate HGM-only vs LGGM-only (currently joint 1.06% gain on WHU-CD: 91.93%→92.99%).

Q6: Generalization Capability Can the method handle semantic change detection (multi-class changes)? Test stronger backbones (Swin Transformer, ResNet101) and verify dual-branch effectiveness. Critically: does DSP work with minimal foreground samples (e.g., 100 Ddsp samples)? This determines real-world applicability.

---

### Note · Authors · 2026-02-01

I have read and agree with the venue's withdrawal policy on behalf of myself and my co-authors.

---

### Meta-Review · Area_Chair_absW · 2026-01-06

**Summary:**

The reviewers identified several critical issues with the paper, including a crude dataset construction strategy for domain-specific pretraining (lacking change-ratio thresholds), a weak theoretical foundation for the entropy-guided mechanism (e.g., unjustified use of entropy difference and sensitive hyperparameters), incomplete comparisons with recent state-of-the-art methods (e.g., omitting Transformer-based approaches), insufficient computational cost analysis, lack of testing in extreme scenarios (e.g., seasonal variations), subjective module evaluations, and inadequate engagement with related work (e.g., similarities to side-tuning and existing decoders). These concerns highlight gaps in novelty, robustness, and practical deployment.

**Reviewer Concerns:**

It might address measurable issues, such as computational cost, by reporting FLOPs or inference time, and dataset construction by adding threshold-based filtering ablations. However, fundamental concerns such as the theoretical justification for EGASM, comprehensive SOTA comparisons, and novelty claims (e.g., versus change-aware pretraining or side-tuning) would likely remain outstanding due to their depth, along with the need for objective module evaluations and broader generalization tests.

**Reviewer Scores:**

No change in the scores. Reviewer oDHG, emphasizing novelty gaps, would likely maintain the score at 2, as related work deficiencies are hard to address rebuttal.

---

### Decision · Program_Chairs · 2026-01-26

Reject